# Magnolol as a Potential Anticancer Agent: A Proposed Mechanistic Insight

**DOI:** 10.3390/molecules27196441

**Published:** 2022-09-29

**Authors:** Xiaofeng Wang, Qingqing Liu, Yuanfeng Fu, Ren-Bo Ding, Xingzhu Qi, Xuejun Zhou, Zhihua Sun, Jiaolin Bao

**Affiliations:** 1Department of Otolaryngology-Head and Neck Surgery, The First Affiliated Hospital of Hainan Medical University, Haikou 570102, China; 2Key Laboratory of Tropical Biological Resources of Ministry of Education, School of Pharmaceutical Sciences, Collaborative Innovation Center of One Health, Hainan University, Haikou 570228, China; 3State Key Laboratory of Quality Research in Chinese Medicine, Institute of Chinese Medical Sciences, University of Macau, Macao 999078, China; 4State International Joint Research Center for Animal Health Breeding, Key Laboratory of Control and Prevention of Animal Disease of Xinjiang Production & Construction Corps, College of Animal Science and Technology, Shihezi University, Shihezi 832003, China

**Keywords:** magnolol, natural products, anticancer, molecular mechanism, signaling pathway

## Abstract

Cancer is a serious disease with high mortality and morbidity worldwide. Natural products have served as a major source for developing new anticancer drugs during recent decades. Magnolol, a representative natural phenolic lignan isolated from *Magnolia officinali*, has attracted considerable attention for its anticancer properties in recent years. Accumulating preclinical studies have demonstrated the tremendous therapeutic potential of magnolol via a wide range of pharmacological mechanisms against cancer. In this review, we summarized the latest advances in preclinical studies investigating anticancer properties of magnolol and described the important signaling pathways explaining its underlying mechanisms. Magnolol was capable of inhibiting cancer growth and metastasis against various cancer types. Magnolol exerted anticancer effects through inhibiting proliferation, inducing cell cycle arrest, provoking apoptosis, restraining migration and invasion, and suppressing angiogenesis. Multiple signaling pathways were also involved in the pharmacological actions of magnolol against cancer, such as PI3K/Akt/mTOR signaling, MAPK signaling and NF-κB signaling. Based on this existing evidence summarized in the review, we have conclusively confirmed magnolol had a multi-target anticancer effect against heterogeneous cancer disease. It is promising to develop magnolol as a drug candidate for cancer therapy in the future.

## 1. Introduction

Cancer is one of the serious diseases that threatens human life and health. The numbers of new cancer cases and deaths rose to 19.29 million and 9.96 million worldwide in 2020, in which Asia accounted for about the half of total numbers worldwide, with 9.50 million and 5.81 million, respectively [1,2]. Natural products used as traditional medicine have long been widely applied to treat cancer in Asian countries. In modern decades, natural products and their derivatives have served as the main source for developing new anticancer drugs such as paclitaxel, topotecan, and vinorelbine. According to recent statistics reported by Newman and Cragg, there are 39.2% approved anticancer drugs (102 out of 260 approved small molecule drugs) developed from natural products or their derivatives since 1981 [3]. *Magnolia officinalis* (*Magnolia officinalis* Rehd.et Wils) is a plant of the *Magnolia* family. According to the Chinese Pharmacopoeia (version 2020), the dried stem bark, root bark, and branch bark of *Magnolia officinalis* plant are named Houpu and have the medicinal uses of drying Dampness, transforming Phlegm and descending Rebellious Qi. Magnolol (5,5′-diallyl-2,2′-dihydroxybiphenyl) and its structural analogue, honokiol (3,5′-diallyl-4,2′-dihydroxybiphenyl), are the main active ingredients of *Magnolia officinalis* bark extract, which also are the representative compounds of phenolic lignan derived from natural products (Figure 1). Both magnolol and honokiol have been reported to exert various biological properties in recent decades, including anticancer, antioxidative, anti-inflammatory, and many other activities. The unique symmetrical diphenol structure of magnolol plays a crucial role in its bioactivity by facilitating the activation of the interactions with the surface of proteins [4,5]. Since magnolol and honokiol are isomers of biphenyl-type neolignan and share similar chemical structure, and the only difference between two compounds is the position of one hydroxyl group. That is, they exhibit very close biological properties but with slight activity difference. For example, in 2,2′-Azobis (2-amidinopropane) dihydrochloride (AAPH)-induced DNA oxidation, both magnolol and honokiol demonstrated good antioxidant capacities and could trap 1.8 and 2.5 radicals in protecting DNA, respectively [6]. Their activity difference was suggested to be related to the intramolecular hydrogen bond formed between di-ortho-hydroxyl groups, which affected the hydrogen atom in hydroxyl group to be abstracted by radicals [6]. Magnolol and honokiol both have showed good anticancer effects in a number of studies, but are with variable activities in different types of cancer cells. There are already several literature reviews to summarize the anticancer role of honokiol against cancer [7,8,9,10,11,12], but few for magnolol. This paper highlights the potential anticancer effect of magnolol. In recent years, there are a number of studies reporting the anticancer effect of magnolol, which exhibited a wide range of pharmacological mechanisms against cancer, including proliferation inhibition, cell cycle arrest, apoptosis induction, metastasis blockage, angiogenesis inhibition and signaling pathways involving phosphoinositide 3-kinase/protein kinase B/mammalian target of rapamycin (PI3K/Akt/mTOR) signaling, mitogen-activated protein kinase (MAPK) signaling and nuclear factor kappa-B (NF-κB) signaling. In this review, we will summarize the latest advances in the anticancer properties of magnolol and its potential mechanistic actions.

## 2. Search Syntax and Inclusion Criteria

Relevant published studies were identified for the years 2002–2022 by searching PubMed database with general string of “(magnolol) AND (cancer OR tumor OR tumour OR carcinoma)”. The citations used in this study were searched by August 31, 2022. The pre-set criteria for inclusion were: (i) the study investigated the anticancer effect of magnolol, not its analogues or derivatives; (ii) the study attempted to elucidate the underlying anticancer mechanism of magnolol; (iii) the study was pharmacological investigation, not pharmaceutical research (e.g., drug delivery system); (iv) only publication in English was included; and (v) other relevant citations not included in the search syntax were manually searched.

## 3. Magnolol Inhibits Proliferation and Growth

Unlimited proliferation is one of the fundamental cancer characteristics, thus restricting tumor cell proliferation represents a promising direction for developing anticancer therapeutics. In recent years, magnolol has exhibited the antiproliferative property against various cancer types. Magnolol effectively inhibited tumor cell growth in bladder cancer [13,14], breast cancer [15,16], cholangiocarcinoma [17], colon cancer [18,19,20,21,22,23], esophagus cancer [24], fibrosarcoma [25], gallbladder cancer [26], gastric cancer [27], glioblastoma [28,29,30,31,32], leukemia [25], liver cancer [21,33,34,35,36], lung cancer [37,38,39,40], lymphoma [41], melanoma [25,42,43], myeloma [44], oral carcinoma [45,46], osteosarcoma [47], ovarian cancer [48], pancreatic cancer [49], prostate cancer [50,51,52,53], renal cancer [54,55], skin cancer [56,57] and thyroid cancer [58] (Table 1). In terms of treatment efficacy, the in vitro IC_50_ values of magnolol for the majority of cancer types varied from 20 to 100 μM under 24-h treatment. The minimum effective inhibitory dose of magnolol in vitro was 2.4 μM as demonstrated in cancer stem cells enriched from oral squamous cell carcinoma [45]. For in vivo mouse studies, 5 mg/kg magnolol was sufficient to restrict tumor growth in bladder cancer [14], colon cancer [19] and gallbladder cancer [26]. The IC_50_ values of magnolol on different cancer cell lines as well as the effective doses used in tumor-bearing mice were summarized in Table 1. Several cellular signaling have been uncovered to participate in antiproliferative effect of Magnolol against cancer, as summarized in Table 1 and Figure 2, mainly involving with PI3K/Akt/mTOR and MAPK/ERK signaling. Meanwhile, the provocation of cell cycle arrest and apoptosis also demonstrate crucial roles on suppressing tumor growth, they are associated with magnolol-induced MAPK/JNK and MAPK/p38 cascades activation, as well as NF-κB signaling blockage. We introduce them in the following paragraphs.

## 4. Magnolol Inhibits PI3K/Akt/mTOR Signaling

The PI3K/Akt/mTOR signaling has been found to be frequently dysregulated pathways of cancer cell proliferation and survival [59]. PI3K catalyzes the conversion of phosphatidylinositol (3,4)-bisphosphate into phosphatidylinositol (3,4,5)-trisphosphate (PIP3), therefore triggers the phosphorylation of Akt [60], subsequently activates mTOR complex 1 (mTORC1) via phosphorylation [61]. It is known that mTOR controls the expression of many proliferation promoting proteins via phosphorylating the p70S6 kinase (p70S6K) and the eukaryotic translation initiation factor 4E binding protein 1 (4E-BP1) [62]. Many studies have reported that magnolol inhibited tumor cell proliferation through attenuating the PI3K/Akt/mTOR signaling. Magnolol has been observed to diminish the phosphorylation of PI3K, Akt, mTOR, p70S6K and 4E-BP1, and this magnolol-induced inactivation of PI3K/Akt/mTOR signaling seems to be cancer type-independent, across a variety of cancer types covering bladder cancer [14], colon cancer [20], gastric cancer [27], glioblastoma [31], lung cancer [37,40], melanoma [43], ovarian cancer [48], prostate cancer [50] and skin cancer [57] (Table 1 and Figure 2). The PI3K/Akt/mTOR signaling usually is activated by ligand binding tyrosine kinase receptors, and PTEN negatively regulates the cascade. Magnolol inactivated the tyrosine kinase receptor EGFR and VEGFR2 [14,50] as well as upregulating PTEN activity, thereby abolishing the abnormal PI3K/Akt/mTOR signaling in cancer cells [38,58]. It is considered that the PI3K/Akt/mTOR cascade serves as a major antiproliferative signaling pathway responsible for magnolol-induced tumor growth inhibition.

## 5. Magnolol Inhibits MAPK Signaling

The MAPK signaling pathway is an important signal transduction pathway that regulates many cellular processes, and its dysregulation leads to aberrant control of cell proliferation, survival and apoptosis [63]. There are three main classical MAPK cascades involving MAPK/extracellular signal-regulated kinase (ERK), MAPK/C-Jun N-Terminal Kinase (JNK) and MAPK/p38 mitogen-activated protein kinase (p38) [64]. All three of these cascades have been reported to be targeted by magnolol to exert anticancer properties as revealed by multiple studies [13,18,22,23,24,30,34,35,36,37,41,43,46,48,56,57], however whether the modulating effect of magnolol on different MAPK cascades is upregulated or downregulated remains controversial (Table 1 and Figure 2). Magnolol could activate MAPK/ERK cascade by increasing the phosphorylation of ERK to provoke and apoptosis and cell cycle arrest in bladder cancer [13], colon cancer [18], esophagus cancer [24], glioblastoma [30], lymphoma [41] and skin cancer [56]. The similar programmed cell death was also observed mediating by magnolol-induced MAPK/JNK and MAPK/p38 activation in bladder cancer [13], esophagus cancer [24], glioblastoma [30], lung cancer [37] and oral carcinoma [46]. However, the MAPK signaling pathway could play a dual role in regulating cancer inhibition process. As reported by other studies, magnolol was capable of hindering NF-κB activation and cell survival through the downregulation of MAPK/ERK cascade [23,34,36,43,48,57]. Additionally, magnolol was also found to block cancer migration and invasion via suppressing ERK and p38 cascades [22,23,34,48]. Therefore, how MAPK cascades acts on magnolol-induced anticancer process remains controversial, it might be due to the reason that the MAPK pathway is predominantly stimulated by various stress signaling and plays a central role in regulating diverse downstream.

## 6. Magnolol Inhibits NF-κB Signaling

NF-κB signaling plays a fundamental role in cancer development and progression, its aberrant activation has been found to be associated with multiple pathways in malignant tumors, including apoptosis inhibition, survival promotion and angiogenesis induction [65,66]. NF-κB activation also triggers epithelial-mesenchymal transition (EMT) to promote cancer migration and invasion [67]. The classic form of NF-κB is the heterodimer of the p50 and p65 subunits, which is controlled by its biological inhibitor IκBs by modulating the translocation of NF-κB from cytoplasm to nucleus. NF-κB is activated by the phosphorylation-induced degradation of IκBs, which facilitates the dissociation of NF-κB from IκB family proteins in cytoplasm [67]. In various cancer types, magnolol exerted the inhibitory effect on NF-κB activity, which is demonstrated by studies of breast cancer [15], cholangiocarcinoma [17], colon cancer [23], liver cancer [34,36], lung cancer [39], myeloma [44], ovarian cancer [48] and skin cancer [57] (Table 1 and Figure 2). For the majority of cases, the inactivation of NF-κB signaling induced by magnolol was to block the phosphorylation of IκBα and/or p65 subunit of NF-κB. Followed by direct NF-κB activity inactivation, magnolol also affected the expression of NF-κB targeting genes involved in apoptosis (Survivin, XIAP, c-FLIP and Mcl-1) [23,34,36], cell cycle (cyclin D1) [17,23,36], metastasis (MMP2, MMP7 and MMP9) [17,23,34,36,44], angiogenesis (VEGF) [23] and inflammation (COX-2) [57]. In general, the magnolol-affected NF-κB signaling inhibition plays crucial functions on regulating tumor growth and metastasis. In particular, it is considered as the primary anti-metastatic signaling of magnolol [15,17,34,44]. 

## 7. Magnolol Induces Cell Cycle Arrest

Cell cycle is tightly regulated by organized checkpoints to maintain normal cell division and proliferation. Many key checkpoint regulators of cell cycle have been identified, which can be classified as three main categories: cyclin, cyclin-dependent kinase (CDK) and cyclin-dependent kinase inhibitor (CKI). They coordinate with each other to constitute a complex regulatory network in the cell cycle. There are four major cyclins involved in cell cycle control, including cyclin A, B1, D1 and E. Entry into the S phase from G1 phase requires cyclin E and D1 activation, cyclin A facilitates the progression through S phase. Cyclin B1 is an essential cyclin to bring cells into mitosis (M) stage. The cyclins form a complex with its corresponding CDKs (CDK1, 2, 4 or 6) to promote cell cycle progression, whereas CKIs, such as p15, p16, p21 and p27, negatively regulate the cell cycle through competing with corresponding cyclins for the binding to CDKs [68]. Abnormal cell cycle regulatory protein expression would lead to unlimited cell replication, which is a hallmark of cancer [69]. Magnolol has been shown to induce cell cycle arrest, consequently leading to apoptosis and cell death in multiple cancer types. Interestingly, it was found that magnolol could trigger cell cycle arrest at three different phases involving sub-G1 [27,34,39,46,55], G0/G1 [13,17,18,21,26,29,30,35,43,47,52,54] and G2/M [13,16,40,56] (Table 1 and Figure 2). Their difference seemed to be magnolol dose-dependent and cancer type-related. As demonstrated in bladder cancer, magnolol at low dose (20 and 40 μM) could induce cell cycle arrest at G0/G1 phase, only when the dose increasing to 60 μM, cell cycle of G2/M phase started to be arrested [13]. Consistent with this finding, the majority of magnolol doses which could induce G0/G1 cell cycle arrest were at the range of 10–40 μM in various other cancer types, including cholangiocarcinoma [17], colon cancer [21], gallbladder cancer [26], glioblastoma [29], liver cancer [35], melanoma [43], osteosarcoma [47] and renal cancer [54]. By comparison, the doses of magnolol trigging G2/M arrest were much higher, with 75–100 μM in skin cancer and 80–100 μM in breast cancer [16,56]. In addition, magnolol could also induce cell cycle arrest at sub-G1 phase in gastric cancer [27], liver cancer [34], lung cancer [39], oral cancer [46] and renal cancer [55]. There was no specific effective dose preference for sub-G1 arrest induction found from these studies, indicating the cancer type difference, mainly caused by genetic background heterogeneity, might also contribute to important influence on how magnolol regulates cell cycle arrest.

## 8. Magnolol Induces Apoptosis

Apoptosis is a strictly programmed cell death process that plays a critical role in physiological and pathological activities. Resisting cell death is a hallmark of cancer development and progression [69]. Therefore, promoting cancer cell death by triggering apoptosis has become a feasible strategy for anticancer therapy. Apoptotic pathways generally can be divided by intrinsic (mitochondrial-mediated) and extrinsic (death receptor-mediated) pathways. The intrinsic pathway is activated by cytochrome c released from mitochondria, which is regulated by the Bcl-2 family members and caspases. The Bcl-2 family includes both pro-apoptotic (e.g., Bax, Bak, Bid, Bad and Bim) and anti-apoptotic (e.g., Bcl-2, Bcl-x and Bcl-XL) proteins [70,71,72]. The intrinsic pathway is usually initiated by intracellular stress stimuli and requires the activation of caspase-9. The extrinsic pathway is initiated by the activation of death receptors, such as the receptors of FASL, TNF, and TRAIL, which are associated with procaspase-8 through the dimerization of the death effect domain, leading to autocatalytic activation of procaspase-8 and triggering apoptosis [73]. Both intrinsic and extrinsic apoptosis induce cleavage caspase family, such as caspase-3, caspase-6 and caspase-7. The apoptotic effects of magnolol have been widely investigated in many studies, meanwhile different cellular mechanisms elucidating magnolol-triggered apoptosis have been explored. The major apoptotic process induced by magnolol was initiated by the mitochondria-dependent intrinsic pathway. Magnolol induced intrinsic apoptosis in various types of cancer (Table 1 and Figure 2), including breast cancer [16], colon cancer [20], glioblastoma [31], liver cancer [34,35,36], lung cancer [37,39,40], lymphoma [41], melanoma [42], osteosarcoma [47], ovarian cancer [48], and thyroid cancer [58]. In general, magnolol significantly decreased Bcl-2 expression and increased Bax expression, leading to an upregulation of the Bax/Bcl-2 ratio; it also induced the release of cytochrome c, increased PARP cleavage, and consequently induced apoptosis in cancer cells. Several studies had found that magnolol-induced intrinsic apoptosis via cleaved-caspase-9, for example, in glioblastoma [31], liver cancer [36], lung cancer [40] and renal cancer [54]. The activated caspase-9 induced the cleavage of downstream caspases-3, which subsequently induced apoptosis. In addition, magnolol also induced the mitochondria to nucleus translocation of apoptosis-inducing factor (AIF) and resulted in apoptosis [16,37]. In addition to the intrinsic apoptosis, magnolol could also provoke the extrinsic apoptotic pathway in fibrosarcoma [25], glioblastoma [28], leukemia [25], melanoma [42], renal cancer [55] and skin cancer [56] (Table 1 and Figure 2). Magnolol was able to induce death receptor mediated caspase-8 activation in a concentration-dependent manner, subsequently activated caspase-3 and PARP by cleavage and resulted in apoptosis [28,55,56]. Furthermore, the downregulated expression of the anti-apoptotic protein Mcl-1, c-FLIP, XIAP and the upregulated expression of pro-apoptotic protein Apaf-1, ATF4 also have been demonstrated to involve in magnolol-induced apoptosis [33,34,47,54,55]. Interestingly, it was found that magnolol could simultaneously activate both intrinsic mitochondria-dependent and extrinsic death receptor-mediated apoptotic pathways against cancer, such as in colon cancer [23], esophagus cancer [24], glioblastoma [28], liver cancer [33], melanoma [42] and oral carcinoma [46]. The intrinsic apoptotic signaling transduction (the loss of mitochondrial membrane potential and cleaved caspase-9) and the extrinsic apoptotic signaling transduction (the activation of FAS, FASL, DR-4, DR-5 and cleaved caspase-8) could be significantly triggered by magnolol at a same time [28,55].

## 9. Magnolol Inhibits Migration and Invasion

Metastasis is the main lethal factor in cancer patients. It involves a sequential cascade, including the migration and invasion of regional tumor cells into neighboring tissues, intravasation into blood or lymphatic vessels, and extravasation into distant organs [74]. To degrade extracellular matrix (ECM) of surrounding tissues and transform tumor cells into invasive mesenchymal phenotype are essential processes involved in cancer cell migration and invasion [75]. Matrix metalloproteinases (MMPs) are calcium endopeptidases that promote ECM degradation [76]. Overexpression of MMPs has been found to be highly associated with tumor metastasis progression [77]. EMT is a critical process to make epithelial tumor cells obtaining the features of invasiveness [78]. The EMT processes manifest as upregulation of mesenchymal markers (α-SMA, fibronectin, N-cadherin, Snail, Slug, Twist and Vimentin) and downregulation of epithelial markers (E-cadherin, cytokeratin, and occludin) [79]. Various studies have reported that magnolol had the capability to suppress cancer cell migration and invasion in different cancer types, including breast cancer [15], cholangiocarcinoma [17], colon cancer [19,20,22,23], esophagus cancer [24], glioblastoma [28,32], liver cancer [33,34,35], myeloma [44], oral carcinoma [45], osteosarcoma [47], ovarian cancer [48], pancreatic cancer [49], prostate cancer [53], and renal cancer [54] (Table 1). These studies proved that magnolol inhibited migration and invasion by decreasing the expressions and activities of MMPs, such as MMP-2, MMP-7, MMP-9, and urokinase-type plasminogen activator (uPA) [15,17,19,23,24,28,33,34,44]. Magnolol could remarkably block EMT by downregulating Vimentin, N-Cadherin, Twist, Slug, Snail and upregulating E-cadherin [19,22,32,49]. A number of studies indicated that several upstream signaling cascades were involving in the modulation of MMPs and EMT by magnolol. Magnolol diminished NF-κB activation by downregulating phosphorylation of p65 and IκBα, which was considered as the primary anti-metastatic signaling of magnolol as evidenced in multiple cancer types [15,17,34,44]. Magnolol also inhibited transforming growth factor beta 1 (TGF-β1)/Smad signaling [22,49], β-Catenin/transcription factor (TCF) signaling [19], protein kinase C delta (PKCδ)/signal transducer and activator of transcription 3 (STAT3) signaling [28]. Additionally, magnolol decreased focal adhesion-related protein expression through attenuating focal adhesion kinase (FAK) and paxillin phosphorylation [32].

## 10. Magnolol Suppresses Angiogenesis

Angiogenesis, the formation of new blood vessels, is critical for supplying nutrients and oxygen required for tumor progression [80]. Tumor angiogenesis is a complex process in which tumor cells interact with endothelial cells. Vascular endothelial growth factor (VEGF) has been described as an essential angiogenic factor that plays a fundamental role in new blood vessel generation [10]. Secreted VEGF interacts with its specific receptors (VEGFR) to initiate endothelial cell proliferation, migration, and colonization, thus to promote angiogenesis. Intratumoral hypoxia is believed to be a main cause of angiogenesis that is closely associated with tumor growth and metastasis [81,82]. Under hypoxic condition, hypoxia-inducible factor 1α (HIF-1α) has been regarded as the primary regulator facilitating angiogenesis by upregulating pro-angiogenic genes in tumor cells, such as VEGF [83]. Thus, the inhibition of hypoxia-mediated tumor angiogenesis is considered a feasible approach to attenuating cancer progression. Magnolol has been found to have effective antiangiogenic activities (Table 1). Specifically, magnolol suppressed the proliferation and tube formation of human umbilical endothelial cells (HUVECs) and mouse embryonic stem (MES)/embryoid body (EB)-derived endothelial-like cells [14,84,85] and inhibited vascular branch formation in chicken chorioallantoic membrane (CAM) [14]. Similar anti-angiogenic evidence was also proved by in vivo studies that magnolol blocked vascular outgrowth in bladder cancer xenografts and subsequently restrained tumor growth [14]. In endothelial cells, magnolol targeted VEGFR-dependent angiogenesis via repressing PI3K/Akt/mTOR and MAPK signaling pathway and inducing ROS-mediated apoptosis [85]. On the other hand, magnolol also acted as an important role in tumor cells through reducing hypoxia /HIF-1α-mediated VEGF secretion [14]. Hypoxia-associated HIF-1α was the primary transcription factor of VEGF. In bladder cancer cells, the hypoxia-induced HIF-1α activation and VEGF secretion were abolished by magnolol through suppressing VEGFR2/PI3K/Akt/mTOR/p70S6K/4E-BP1 cascade [14]. Meanwhile, the magnolol-mediated inhibition of HIF-1α protein expression is associated with increased HIF-1α protein degradation and protein synthesis blockage [14].

## 11. Synthetic Analogues of Magnolol and Their Anticancer Activities

To enhance the anticancer efficacy of magnolol, several structural modification studies have been investigated (Figure 3). Magnolol analogue 5,5′-dibutyl-3,3′-diiodo-[1,1′-biphenyl]-2,2′-diol (named as Ery5, (a)) exhibited strong antiproliferative activity with over 20-fold increase against HL-60 and PC-3 cancer cell lines compared with magnolol [86]. Ery5 could effectively induce cell cycle arrest at Sub-G1, trigger autophagic cancer cell death and inhibit angiogenesis [86,87,88]. Xu et al. synthesized a serious of magnolol-based Mannich base derivatives by introducing Mannich base groups at the ortho position of the phenolic hydroxyl at C-2′ [89]. Most of these magnolol derivatives displayed more potent antiproliferative activity against cancer cells than magnolol. Among them, 5,5′-diallyl-3-(morpholino(3,4,5-trimethoxyphenyl)methyl)-[1,1′-biphenyl]-2,2′-diol (compound 3p, (b)) had the most potent anticancer activity against T47D cancer cells with 76.1-fold enhancement of cytotoxic effect compared with magnolol [89]. Tang et al. introduced nitrogen heterocycle at C-3 position of magnolol in order to improve the water solubility and antitumor activity [90]. Magnolol analogue 5,5′-diallyl-3-((4-aminopiperidin-1-yl)methyl)-[1,1′-biphenyl]-2,2′-diol (compound C2, (c)) exhibited significant and broad-spectrum antiproliferative activity with approximately 8-fold more potent than magnolol in NSCLC cell lines [90]. Based on the core of compound C2, Zhao et al. continued to synthesize 5,5′-diallyl-3-((4-aminopiperidin-1-yl)methyl)-2′-isopropoxy-[1,1′-biphenyl]-2-ol (compound 30, (d)) with a piperityl group substituting the hydrogen atom at C-3 position of magnolol, which exhibited potent antiproliferative activities on H460, HCC827 and H1975 cancer cell lines with approximately 100-fold more potent than magnolol [91]. Chen et al. synthesized magnolol analogue 2-(4-((2-((5′,6-diallyl-2′-hydroxy-[1,1′-biphenyl]-3-yl)oxy)ethyl)sulfinyl)butyl)isoindoline-1,3-dione (compound CT2-3, €), which showed significant cytotoxicity against A549 and H460 cancer cell lines [92]. But the antiproliferative activity of compound CT2-3 was not as strong as the analogues synthesized by other research groups, only 2-fold better than magnolol [92]. Tao et al. synthesized a magnolol-sulforaphane hybrid 2′,5-diallyl-5′-(2-((4-isothiocyanatobutyl)sulfinyl)ethoxy)-[1,1′-biphenyl]-2-ol (compound CT1-3, (f)), exhibiting at least 5-fold more efficient anticancer activity than magnolol in multiple cancer cell lines [93]. More recently, Sun et al. designed and synthesized a series of magnolol derivatives showing better cytotoxic activity compared to magnolol [94]. Among them, 5,5′-diallyl-2′-((4-fluorobenzyl)oxy)-[1,1′-biphenyl]-2-ol (compound 6a, (g)) was the most potent one to inhibit cancer cell proliferation, migration and invasion. The results suggested that substitution of the benzyl group with F atoms at the C2 position on magnolol was a viable strategy for structural optimization [94].

## 12. Conclusions and Prospects

Cancer is a serious malignant disease with high morbidity and mortality worldwide. As far as we know, the complexity of tumor heterogeneity brings cancer to be a multifactorial disease, which involves in various signaling pathways during tumor initiation, progression and recurrence [95]. Therefore, to develop new drugs with multiple targets against cancer has become a promising and popular trend. Chinese herbal medicines as anticancer drug discovery sources have attracted increasing attentions in recent years because they have been demonstrated with advantages of evidence-based effectiveness, multi-target therapy, and few side effects. Accumulating preclinical studies have proved the tremendous anticancer potential of magnolol, a representative compound derived from Chinese herbal medicines, as summarized in the present review. Magnolol exerted anticancer effects through various molecular mechanisms, including proliferation, cell cycle, apoptosis, metastasis, angiogenesis and so on. Multiple signaling pathways were also involved in the pharmacological mechanisms of magnolol against cancer, such as PI3K/Akt/mTOR signaling, MAPK signaling and NF-κB signaling (Table 1 and Figure 2). Based on this evidence, the multi-target anti-cancer properties of magnolol have been initially confirmed as supporting its druggable potential against heterogeneous cancer. Furthermore, unlike chemotherapy, either magnolol or *Magnolia* bark extract had no known toxicities and side effects reported so far, which makes this natural agent a potentially safe drug candidate. In addition, magnolol has the ability to successfully cross the blood-brain barrier and rapidly distribute into different brain regions [96], which expands its pharmacological potential against brain tumors. However, it is worth noting that the water solubility, gastrointestinal absorption and first-pass metabolism of magnolol are poor, with oral bioavailability less than 10% [97], which restricts its clinical application. Despite the promising results on anticancer activity of magnolol, further research is required to improve its bioavailability, including to investigate appropriate liposomal formulation, nanocarrier delivery system and chemical structure modification to enhance magnolol in vivo efficacy [98,99,100]. And so far, no clinical evidence has been obtained to show the efficacy of magnolol. To fully realize the therapeutic potential of magnolol, more clinical investigations are necessary. Nevertheless, based on existing preclinical studies, we have reason to believe that magnolol may potentially be developed as a promising natural agent to fight against cancer. To sum up, we have reviewed the anticancer effects of magnolol against various cancer types and elucidated its main underlying pharmacological mechanisms, including proliferation inhibition, cell cycle arrest, apoptosis induction, metastasis blockage, angiogenesis inhibition, and pathways involving in PI3K/Akt/mTOR, MAPK and NF-κB signaling (Table 1 and Figure 2). The preclinical studies on magnolol are intriguing and suggest it as a promising multi-target drug candidate for cancer therapy.

## Figures and Tables

**Figure 1 molecules-27-06441-f001:**
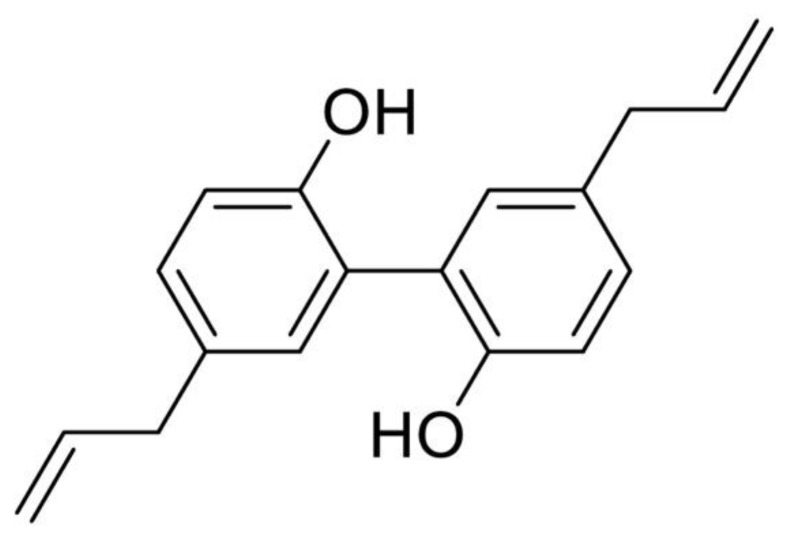
Chemical structure of magnolol.

**Figure 2 molecules-27-06441-f002:**
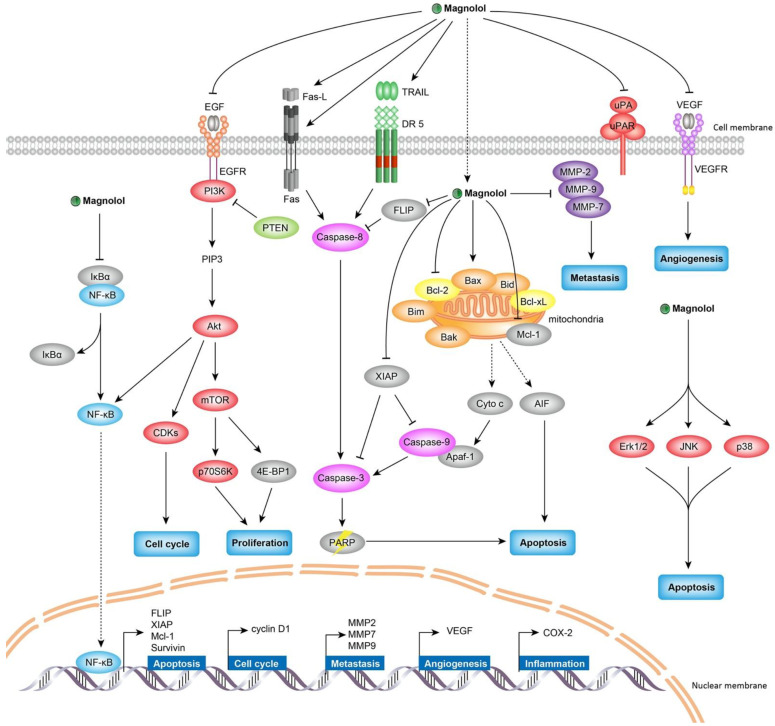
The anticancer mechanism of magnolol. Magnolol exerts anticancer effects through inhibiting proliferation, inducing cell cycle arrest, provoking apoptosis, restraining migration and invasion, and suppressing angiogenesis. Multiple signaling pathways are involved in the pharmacological actions of magnolol against cancer, such as PI3K/Akt/mTOR signaling, MAPK signaling, and NF-κB signaling.

**Figure 3 molecules-27-06441-f003:**
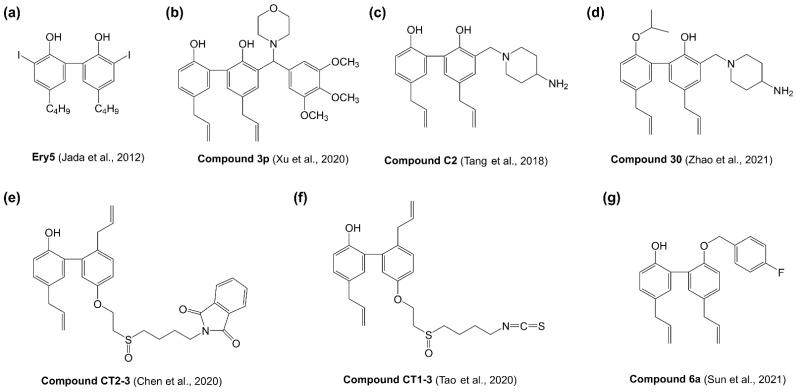
The structure of magnolol synthetic analogues. (**a**) 5,5′-dibutyl-3,3′-diiodo-[1,1′-biphenyl]-2,2′-diol; (**b**) 5,5′-diallyl-3-(morpholino(3,4,5-trimethoxyphenyl)methyl)-[1,1′-biphenyl]-2,2′-diol; (**c**) 5,5′-diallyl-3-((4-aminopiperidin-1-yl)methyl)-[1,1′-biphenyl]-2,2′-diol; (**d**) 5,5′-diallyl-3-((4-aminopiperidin-1-yl)methyl)-2′-isopropoxy-[1,1′-biphenyl]-2-ol; (**e**) 2-(4-((2-((5′,6-diallyl-2′-hydroxy-[1,1′-biphenyl]-3-yl)oxy)ethyl)sulfinyl)butyl)isoindoline-1,3-dione; (**f**) 2′,5-diallyl-5′-(2-((4-isothiocyanatobutyl)sulfinyl)ethoxy)-[1,1′-biphenyl]-2-ol; (**g**) 5,5′-diallyl-2′-((4-fluorobenzyl)oxy)-[1,1′-biphenyl]-2-ol.

**Table 1 molecules-27-06441-t001:** The anticancer effect of magnolol against various cancer types.

No.	Cancer Types	Cell Lines Used	Models	IC_50_ In Vitro/ Drug Dose In Vivo	Mechanism of Action	Reference
1	Bladder cancer	5637	In vitro	60 μM (24 h)	Induces cell cycle arrest at G0/G1 (low dose) and G2/M (high dose) Cyclin D1/E/B1, CDK2/4↓ p-Cdc25c, p-Cdc2↓; p27↑ Activates p38/MAPK signaling p-ERK, p-p38, p-JNK↑	[13]
2	Bladder cancer	T24	In vitro	NA	Inhibits hypoxia-induced angiogenesis HIF-1α, CD31, CA-IX↓ Inhibits VEGF/PI3K/Akt/mTOR signaling VEGF, p-VEGFR2↓ p-Akt, p-mTOR, p-p70S6K, p-4E-BP1↓	[14]
			In vivo	i.p. 5–10 mg/kg every day		
3	Breast cancer	MCF-7	In vitro	58.27 μM (24 h); 53.39 μM (48 h); 49.56 μM (72 h)	Induces cell cycle arrest at G2/M p21, p53↑; CDK1, Cyclin B1↓ Induces apoptosis Bax, Cytochrome C, AIF, Cleaved PARP↑ Bcl-2↓	[16]
4	Breast cancer	MDA-MB-231; MDA-MB-468; MDA-MB-453; MDA-MB-435S; MCF-7; SK-BR3	In vitro	MDA-MB-231: 25.32 μM (24 h) MDA-MB-468: 24.79 μM (24 h) MDA-MB-453: 35.69 μM (24 h) MDA-MB-435S: 25.49 μM (24 h) MCF-7: 70.52 μM (24 h) SK-BR3: 59.4 μM (24 h)	Inhibits migration and invasion MMP-2/9↓ Inhibits NF-κB signaling NF-κB activity, p-p65, p-IκBα↓	[15]
		MDA-MB-231; MCF-7	In vivo	i.p. 40 mg/kg four times a week		
5	Cholangiocarcinoma	QBC939; SK-ChA-1; MZ-ChA-1; RBE	In vitro	QBC939: 40–80 μM (24 h); ~40 μM (48 h); 20–40 μM (72 h) SK-ChA-1: 40–80 μM (24 h); 40–80 μM (48 h); 20–40 μM (72 h) MZ-ChA-1: 40–80 μM (24 h); ~40 μM (48 h); 20–40 μM(72 h) RBE: 80–100 μM (24 h); 40–80 μM (48 h); 40–80 μM (72 h)	Induces cell cycle arrest at G0/G1 Cyclin D1↓ Inhibits migration and invasion MMP-2/7/9↓ Inhibits NF-κB signaling p-IκBα, p-p65↓	[17]
		MZ-ChA-1	In vivo	i.p. 40 mg/kg every day		
6	Colon cancer	COLO-205	In vitro	NA	Induces cell cycle arrest at G0/G1 p21, p-ERK, p-Raf, ↑	[18]
7	Colon cancer	SW480; HCT116	In vitro	SW480: 166.9 μM (24 h); 67.9 μM(48 h); 53.5 μM (72 h) HCT116: 95.6 μM (24 h); 75.4 μM (48 h); 72.9 μM (72 h)	Inhibits β-Catenin/TCF signaling β-Catenin, c-Myc, Cyclin D1↓ Inhibits migration and invasion E-cadherin↑; MMP-7 and uPA↓	[19]
		HCT116	In vivo	i.p. 5 mg/kg three times a week		
8	Colon cancer	HCT116	In vitro	25–50 μM (24 h)	Induces apoptosis Cleaved Caspase-3, Cleaved PARP↑ Bax, p53, Cytochrome C↑; Bcl-2↓ Activates AMPK signaling p-AMPK↑ Inhibits mTOR signaling p-mTOR, p-p70S6K↓ Inhibits migration and invasion	[20]
9	Colon cancer	COLO-205; HT29	In vitro	NA	Induces cell cycle arrest at G0/G1 P21↑; CDK2, Cyclin A/E↓	[21]
		COLO-205	In vivo	i.p. 100 mg/kg 5 times a week		
10	Colon cancer	HCT116; SW480	In vitro	NA	Inhibits migration and invasion E-Cadherin, ZO-1, Claudin↑ N-Cadherin, Twist, Slug, Snail↓ TGF-β, TGF-β RI↓ p-ERK, p-GSK3β, p-Smad↓	[22]
11	Colon cancer	CT26; HT29	In vitro	75 μM (24 h)	Inhibits NF-κB signaling NF-κB activity, p-p65↓ Inhibits PKCδ/MAPK/Akt signaling p-PKCδ, p-ERK, p-Akt↓ Induces apoptosis Cleaved Caspase-3/8/9, FAS, FASL↑ Mcl-1, c-FLIP, XIAP↓ Inhibits migration and invasion MMP-2/9, uPA↓ Cyclin D1↓	[23]
			In vivo	50–100 mg/kg twice a week		
12	Esophagus cancer	TE-1; KYSE-150; Eca-109	In vitro	KYSE-150: 50–100 μM (24 h); ~50 μM (48 h) Eca-109: 50–100 μM (24 h); 50–100 μM (48 h); TE-1: 100–150 μM (24 h); 100–150 μM (48 h);	Induces apoptosis Cleaved Caspase-3/8/9, Bax↑; Bcl-2↓ Inhibits migration MMP-2↓ Activates p38/MAPK signaling p-p38, p-ERK↑	[24]
		KYSE-150	In vivo	i.p.30mg/kg every other day		
13	Fibrosarcoma	HT1080	In vitro	NA	Induces apoptosis Caspases-3/8 activity↑	[25]
14	Gallbladder cancer	GBC-SD SGC996	In vitro	GBC-SD: 20.5 μM (48 h) SGC996: 14.9 μM (48 h)	Induces apoptosis Cleaved Caspase-3/9, Cleaved PARP↑ Bax, p53, p21↑; Bcl-2↓ Induces cell cycle arrest at G0/G1 Cdc25A, CDK2, Cyclin D1↓	[26]
		GBC-SD	In vivo	i.p. 5–20 mg/kg every day		
15	Gastric cancer	SGC-7901	In vitro	50–100 μM (48 h)	Induces cell cycle arrest at Sub-G1 and S Induces apoptosis Cleaved Caspase-3, Bax↑; Bcl-2↓ Inhibits PI3K/Akt signaling p-PI3K, p-Akt↓	[27]
16	Glioblastoma	GBM8401; BP5	In vitro	GBM8401: 25 μM(48 h) BP5: 150 μM(48 h)	Induces apoptosis Cleaved Caspase-3/8/9, FAS, FASL, DR4/5↑ Inhibits PKCδ/STAT3 signaling p-STAT3, p-PKCδ↓ Inhibits migration and invasion MMP-9, uPA↓	[28]
17	Glioblastoma	U373	In vitro	NA	Induces cell cycle arrest at G0/G1 Cyclin A/D1↓; p21↑	[29]
18	Glioblastoma	U373	In vitro	NA	Induces apoptosis Induces cell cycle arrest at G0/G1 P21, p27↑ Activates cSrc/MAPK signaling p-cSrc, p-ERK, p-p38, p-Akt↑	[30]
			In vivo	i.p. 100 mg/kg Every other day		
19	Glioblastoma	U87MG; LN229; GBM8401	In vitro	NA	Induces apoptosis Cleaved Caspase-3↑; Bcl-2↑ Inhibits PI3K/Akt signaling p-PI3K, p-Akt↓	[31]
		LN229	In vivo	i.p. 20 mg/kg every day		
20	Glioblastoma	U87MG; LN229	In vitro	NA	Inhibits migration and invasion p-FAK, p-paxillin, integrin β1/β3, p-Src↓ p-MLC, p-MLCK, p-MYPY1↓ N-Cadherin, β-catenin↓	[32]
		LN229	In vivo	i.p. 20 mg/kg every day		
21	Leukemia	THP-1	In vitro	NA	Induces apoptosis Caspases-3/8 activity↑	[25]
22	Liver cancer	HepG2; Hep-3B	In vitro	NA	Induces cell cycle arrest at G0/G1 P21↑; CDK2, Cyclin A/E↓	[21]
23	Liver cancer	Hep-3B; SK-Hep1	In vitro	Hep-3B: 75–100 μM (48 h) SK-Hep1: 75–100 μM (48 h)	Induces apoptosis Cleaved Caspase-3/8/9, Cleaved PARP↑ Bax, Bak, FAS↑ XIAP, C-FLIP, Mcl-1, MDC-1↓ Inhibits migration and invasion MMP-9, VEGF-A↓	[33]
24	Liver cancer	SK-Hep1	In vitro	~150 μM (48 h)	Induces cell cycle arrest at Sub-G1 Induces apoptosis Caspase-3 activity↑; Survivin, XIAP↓ Inhibits migration and invasion MMP-2/9, uPA↓ Inhibits NF-κB signaling NF-κB activity, p-ERK↓	[34]
25	Liver cancer	HepG2	In vitro	~30 μM (48 h)	Induces cell cycle arrest at G0/G1 Inhibits migration and invasion Induces apoptosis Bax, Cleaved PARP, Cytochrome C↑; Bcl-2↓ Activates ER stress signaling GRP78, p-ERK, p-eIF2α, CHOP↑	[35]
			In vivo	i.p. 10–30 mg/kg every day		
26	Liver cancer	SK-Hep1	In vivo	p.o. 50–100 mg/kg every day	Induces apoptosis XIAP, c-FLIP, Mcl-1↓ Cleaved Caspase-3/9↑ Inhibits NF-κB signaling NF-κB activity, p-p65↓ p-ERK, MMP-9, Cyclin D1↓	[36]
27	Lung cancer	A549; H441; H520	In vitro	A549: 80–100 μM (24 h) H441: 80–100 μM (24 h) H520: ~100 μM (24 h)	Induces apoptosis Bid, Bax, Cytochrome C, Cleaved PARP↑ Induce mitochondria-to-nuclear translocation of AIF and EndoG Activates p38/JNK signaling p-p38, p-JNK↑ Inhibits PI3K/Akt signaling p-PI3K, p-Akt, p-ERK↓	[37]
28	Lung cancer	H460	In vitro	80–100 μM (24 h); 60–80 μM (48 h)	Induces autophagy Inhibits PTEN/Akt signaling PTEN↑; p-Akt↓	[38]
29	Lung cancer	A549	In vitro	50–100 μM (24 h)	Induces cell cycle arrest at sub-G1 Induces apoptosis Cleaved PARP↑ Inhibits NF-κB signaling NF-κB activity, p65↓ Inhibits angiogenesis	[39]
30	Lung cancer	A549; H1299	In vitro	A549: 5 μM (72 h) H1299: 5 μM (72 h)	Induces cell cycle arrest at G2/M Cyclin B1, CDK2, p21, P27↑ p-Cdc2, Cdc25c, Cyclin D1/A2, CDK1/4↓ Inhibits microtubule polymerization MPM-2↑ Induces apoptosis Cleaved Caspase-9, Cleaved PARP, Bax↑ Bcl-2↓ Induces autophagy ATG5/12, LC3B-II/I ratio↑ mTOR, p62, p-Akt↓	[40]
		A549	In vivo	i.p. 25 mg/kg every other day		
31	Lymphoma	U937	In vitro	31.63 μM (72 h)	Induces apoptosis Cleaved Caspase-3, AIF, Bax↑ Bcl-2, c-Myc↓ p-ERK↑	[41]
32	Melanoma	A375-S2	In vitro	50–100 μM (24 h); ~50 μM (48 h)	Induces apoptosis Caspase-3/8 activity, Cleaved PARP↑ Bax, Cytochrome C↑; Bcl-2, ICAD↓	[42]
33	Melanoma	WM164; WM1366; HaCaT; D24;	In vitro	NA	Induces cell cycle arrest at G0/G1 Induces apoptosis Inhibits PI3K/Akt signaling p-Akt, p-mTOR, p-ERK↓	[43]
34	Melanoma	B16-BL6	In vitro	NA	Induces apoptosis Caspases-3/8 activity↑	[25]
35	Myeloma	U266; LP1	In vitro	U266: 20–40 μM (24 h); 20–40 μM (48 h) LP1: 20–40 μM (24 h); 10–20 μM (48 h)	Induces apoptosis Inhibits migration and invasion MMP7/9↓ Upregulates miR-129 Inhibits NF-κB signaling p-p65, p-IκBα↓	[44]
36	Oral carcinoma	SAS; GNM; OECM1	In vitro	SAS: 2.4 μM (24 h) GNM: 11.7 μM (24 h) OECM1: 4.1 μM (24 h)	Inhibits cancer stemness ALDH1↓ Inhibits migration and invasion Inhibits STAT3/IL6 signaling p-STAT3, IL6↓	[45]
37	Oral carcinoma	HSC-3; SCC-9	In vitro	HSC-3: 50–75 μM (24 h) SCC-9: 50–75 μM (24 h)	Induces apoptosis Cleaved Caspase-3/8/9, Cleaved PARP↑ HO-1↑; cIAP1↓ Induces cell cycle arrest at Sub-G1 Activates MAPK/ERK signaling p-ERK, p-JNK, p-p38↑	[46]
38	Osteosarcoma	MG-63; 143B	In vitro	MG-63: 32.9 μM (24 h); 27.8 μM (48 h) 143B: 30.3 μM (24 h); 25.1 μM (48 h)	Inhibits migration and invasion Induces cell cycle arrest at G0/G1 P27↑; CDK2, Cyclin D1↓ Induces apoptosis P53, Bax, Cytochrome C, Apaf-1↑ Cleaved Caspase-3/9↑; Bcl-2↓	[47]
39	Ovarian cancer	SKOV3; TOV21G	In vitro	SKOV3: 25–50 μM (48 h); 25–50 μM (72 h) TOV21G: 50–100 μM (48 h); 50–100 μM (72 h)	Inhibits HER2 signaling HER2, p-HER2↓ Inhibits NF-κB signaling p65↓ Inhibits PI3K/Akt/mTOR signaling p-Akt, p-mTOR, p-GSK3β, p38↓ Inhibits migration Induces apoptosis Cleaved Caspase-3, Cleaved PARP↑	[48]
40	Pancreatic cancer	Panc-1; AsPC-1	In vitro	Panc-1: 140.5 μM (24 h); 117.3 μM (48 h); 96.4 μM (72 h) AsPC-1: 160 μM (24 h); 104.2 μM (48 h); 75.4 μM (72 h)	Inhibits migration and invasion Inhibits EMT E-Cadherin↑; Vimentin↓ Inhibits TGF-β1/Smad signaling p-SMAD2/3↓	[49]
		AsPC-1	In vivo	i.p. 50 mg/kg every day		
41	Prostate cancer	LNCaP; PC-3; DU-145	In vitro	LNCaP:53 μM (24 h) PC-3:60 μM (24 h) DU-145:70 μM (24 h)	Induces apoptosis Cleaved Caspase-3/8/9, Cleaved PARP↑ Bax, Cytochrome C↑; p-Bad↓ Inhibits EGFR/PI3K/Akt signaling p-EGFR, p-PI3K, p-Akt, p-PDK1↓	[50]
42	Prostate cancer	LNCaP; PC3	In vitro	NA	Inhibits IGF-1 signaling IGF-1, p-IGR-1R, IGFBP5↓ IGFBP-3/4↑	[51]
43	Prostate cancer	Du145; PC3	In vitro	Du145: ~40 μM (24 h) PC3: ~80 μM (24 h)	Induces cell cycle arrest at G0/G1 CDK2, Cyclin D1↓	[52]
44	Prostate cancer	PC-3	In vitro	~40 μM (24 h); 25–30 μM (48 h); 25–30 μM (72 h)	Inhibits migration and invasion MMP-2/9, COX-1/2↓	[53]
45	Renal cancer	786-O; OS-RC-2	In vitro	786-O: 30.29 μM (24 h); 25.4 μM (48 h) OS-RC-2: 31.19 μM (24 h); 25.9 μM (48 h)	Inhibits migration and invasion Induces cell cycle arrest at G0/G1 P53, p21↑; Cyclin D1, CDK2↓ Induces apoptosis Bax, Cytochrome C, Apaf-1↑ Cleaved Caspase-3/9↑; Bcl-2↓	[54]
46	Renal cancer	Caki-1; ACHN	In vitro	NA	Induces cell cycle arrest at Sub-G1 Induces TRAIL-mediated apoptosis Cleaved Caspase-3, Cleaved PARP, DR5↑ Mcl-1, c-FLIP↓; ATF4↑	[55]
47	Skin cancer	A431	In vitro	NA	Induces apoptosis Cleaved Caspase-3/8, Cleaved PARP↑ Induces cell cycle arrest at G2/M Cyclin A/B1/D1, CDK2/4, Cdc2/25A/25C↓ P21, p-ERK↑ BRAF, p-STAT3, p-MEK, p-Akt↓	[56]
		UVB-induced carcinogenesis	In vivo	topically applied 30–60 μg/mouse 5 times a week		
48	Skin cancer	TP-induced carcinogenesis	In vivo	topically applied 1–5 μM twice a week	Inhibits inflammation iNOS, COX-2↓ Inhibits NF-κB signaling p-p65, p-IκBα↓ Inhibits MAPK signaling p-p38, p-ERK↓ Inhibits PI3K/Akt signaling p-PI3K, p-Akt↓	[57]
49	Thyroid cancer	CGTH W-2	In vitro	NA	Induces apoptosis Cleaved Caspase-3/7, Cleaved PARP↑ Cytochrome C↑ p-PTEN, p-Akt↓	[58]

↑ represents upregulation, ↓ represents downregulation.

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
