# Peer review of "Magnolol as a Potential Anticancer Agent: A Proposed Mechanistic Insight"

_molecules, 2022, doi:10.3390/molecules27196441_

Round 1

Reviewer 1 Report

The manuscript is well written and introduces a new research topic but lacks in several areas; these issues must be addressed to make it interesting for the journal readers.

·    Lacks graphical abstract in the manuscript.

·  There should be a section on Materials and methods. What are the criteria for exclusion and inclusion?

·    What is the timeline of this review?

· Honokiol and magnolol are the main bioactive compounds of M. officinalis. Honokiol, a multi-target compound, showed antitumor activity against liver, colon, pancreatic, and lung cancer cells. Which has been shown better antitumor activity, “Why? Discuss in the introduction section.

Reviewer 2 Report

Reviewers’ comments for the Manuscript ID: Molecules-1934287

The manuscript title:Magnolol as a potential anticancer agent: a proposed mechanistic insight “by Xiaofeng Wang, et al. in the current manuscripts authors comprehensively summarized the anti-cancer effect of small biphenyl based natural product Magnolol involving various signaling pathways, it was well organized and well written manuscript and suitable for publication in “Molecules journal”, after addressing below comments.

General comments

1)      Although authors provided the anticancer effect of magnolol against various cancer types in table 1, but some more additional data discussion in main draft should be provided about cell proliferation, cell cycle arrest, apoptosis, migration, invasion and suppressing angiogenesis as well as each signaling pathways (PI3K, Akt, mTOR, MAPK etc).

2)      In order to improve anticancer efficiency of Magnolol, several synthetic analogues have been reported, so it would be better to summarize the anticancer effects of synthetic analogues in comparison to Magnolol.
